# The Complex Rayleigh Waves in a Functionally Graded Piezoelectric Half-Space: An Improvement of the Laguerre Polynomial Approach

**DOI:** 10.3390/ma13102320

**Published:** 2020-05-18

**Authors:** Ke Li, Shuangxi Jing, Jiangong Yu, Xiaoming Zhang, Bo Zhang

**Affiliations:** School of Mechanical and Power Engineering, Henan Polytechnic University, Jiaozuo 454003, China; like@hpu.edu.cn (K.L.); jsx@hpu.edu.cn (S.J.); zxmworld11@hpu.edu.cn (X.Z.); bozhanghpu@163.com (B.Z.)

**Keywords:** complex Rayleigh waves, functionally graded piezoelectric material, orthogonal function, half-space

## Abstract

The research on the propagation of surface waves has received considerable attention in order to improve the efficiency and natural life of the surface acoustic wave devices, but the investigation on complex Rayleigh waves in functionally graded piezoelectric material (FGPM) is quite limited. In this paper, an improved Laguerre orthogonal function technique is presented to solve the problem of the complex Rayleigh waves in an FGPM half-space, which can obtain not only the solution of purely real values but also that of purely imaginary and complex values. The three-dimensional dispersion curves are generated in complex space to explore the influence of the gradient coefficients. The displacement amplitude distributions are plotted to investigate the conversion process from complex wave mode to propagating wave mode. Finally, the curves of phase velocity to the ratio of wave loss decrements are illustrated, which offers extra convenience for finding the high phase velocity points where the complex wave loss is near zero.

## 1. Introduction

Over the past decades, considerable attention has been focused on the investigation of functionally graded materials (FGMs), because the properties of this kind of heterogeneous composite materials can vary in preferred direction. These materials have been widely used in the thermal-protection systems of aerospace structures, so FGMs attract more and more interest of researchers in materials engineering. In recent years, however, FGPM has received rapt attention, because the material can be manufactured properly [1] with the development of material technology, and has been used in the manufacture of surface acoustic wave devices. For example, J. Qiu et al. [2] successfully used this kind of material to reduce the mechanical damping and heat generation of an actuator in 2003. All these cases show that FGPM is a promising material. On the other hand, surface acoustic wave technique has been adopted normally in signal processing, geophysics, and flow measurement, which makes it ever-increasingly important to study the wave propagation behaviors and characteristics in FGPM structures.

Many ground-breaking studies have been carried out by many researchers to explore the wave propagation behavior in FGPM, and various numerical methods have been proposed at the same time [3]. Early in 2004, Li et al. [4] analyzed the dispersion and anti-dispersion of Love waves in an FGPM half-space using the Wentzel–Kramers–Brillouin (WKB) method. Later, the WKB method was employed to investigate the propagation of Rayleigh surface waves in transversely isotropic FGPM half-space [5], and this method was also applied to study the dispersion relations and the stress distributions in an FGM layered half-space with initial stress [6]. Another approach to deal with the wave propagation in FGPM is the analytical technique. Du et al. [7] obtained the dispersion relations of the Love waves in FGPM layered half-space using an analytical approach, assuming that all material properties have the same exponential function distribution. Qian et al. [8] investigated the effect of gradient coefficient on dispersion relation and phase velocity by studying the characters of transverse surface waves using an analytical technique. The Love waves propagation in the FGPM buffer layer between the upper layer and the substrate was studied using the power series technique [9], and then the method was deployed to investigate the propagation behavior of Lamb waves in functionally graded piezoelectric-piezomagnetic material (FGPPM) plate [10]. Salah et al. combined the ordinary differential equation method with the general stiffness matrix method to investigate the propagation of Love and Rayleigh waves in FGPM half-space (see [11,12]). Very recently, Kong [13] and Mondal et al. [14] studied the propagating characteristics of SH waves in FGPM layered half-space using different method, respectively. On the problem of wave propagation in FGPM plates, C.Othmani et al. proposed a Legendre polynomial method to study the effect of the gradient coefficients on the Lamb phase velocity [15]. Later, they investigated the relationships of volume fraction, gradient coefficient, dispersion curves and phase velocity in a more complex structure of GaAs-FGPM-AlAs sandwich plate [16].

The above-mentioned studies on FGPM were conducted for the propagating waves. However, the studies on complex Rayleigh waves in the FGPM structures are rarely ever found. Zhang et al. [17] mentioned the complex acoustic parts of the Rayleigh wave in a half-space, and Rose [18] gave the explanation of the concept of complex waves, but their further research about the complex Rayleigh waves in FGPM half-spaces cannot be found. As is known, the wave equation can be expressed by angular frequency and wavenumber. When the equation is solved in complex space, it will generate three kinds of values: purely real value, purely imaginary values, or complex values. The real solutions represent propagating modes which do not need explanation anymore, and the solutions of purely imaginary values correspond to non-propagating modes because their amplitude decays exponentially. Nevertheless, it is worth noting that the non-propagating modes, corresponding to the complex values, exhibit the decay following an exponentially damped trigonometric distribution with distance [19]. There exist two big reasons the complex wave modes have been focused on: (1) these complex wave modes will travel a long distance if their imaginary parts are small enough, and they even could be converted to propagating wave modes when some conditions are satisfied; and (2) the complex wave mode can present much higher phase velocity than the surface acoustic wave [20], and further can be introduced to fabricate the SAW devices operating at higher frequencies.

Even though studies on complex Rayleigh waves in FGPM half-space are hardly found, the work on complex propagating waves has been carried out in other structures. It has been proved that some complex Lamb-like waves can be converted to propagating waves [21,22], but no research has been performed regarding the complex Rayleigh waves in FGPM half-space. In 1969, Lim and Farnell [23] began exploring the existence of the wave mode with phase velocity much higher than bulk wave. From then on, it became an ongoing important topic to study this kind of high phase velocity existing in the form of complex wave modes. Boyer et al. [24] solved the difficulty of poor estimation of the electroacoustical admittance for complex surface acoustic wave transducers by using fitting analytical function method. Benetti et al. not only carried out the theoretical investigation of complex surface acoustic wave properties in a layered structure [25], but also verified their results using experiment [26]. More research in this field can be found in the literature [27,28,29,30]. Thus far, most studies of complex surface acoustic waves have been limited in the structure of composite plate or simple layered half-space, and have not involved the structure of FGPM half-space.

Thus, in the present contribution, an improved method based on the Laguerre polynomial technique is proposed to investigate the complex Rayleigh waves in the FGPM half-space. The Laguerre polynomials are widely used to solve wave propagation problems in half-space because they are orthogonal over $\left[0,\infty \right)$. Datta and Hunsinger [31] achieved surface wave solutions in piezoelectric half-space by using the Laguerre orthogonal polynomials while avoiding iterative calculation. Gubernatis and Maradudin [32] investigated the calculation of wave properties in an inhomogeneous half-space using the Laguerre orthogonal polynomials. Kim and Hunt investigated the surface acoustic wave propagation in multilyered half-space by using the Laguerre polynomial technique [33] and extended the method to examine the acoustic fields and velocities for surface acoustic wave propagation [34]. The conventional Laguerre orthogonal polynomial approach takes the wave numbers *k* as independent variables while taking the frequencies $\omega^{2}$ as eigenvalues, but only one square term is related to ω in the Rayleigh wave governing equation. Thus, it can only obtain the real wavenumber solutions of propagating waves. However, the improved method in this paper takes the wave number *k* as eigenvalues while taking ω as independent variables. Both the linear and square terms of *k* exist in its governing equation, so the improved method not only can present the real wave number solution but also can get the solution of purely imaginary and complex wave number. The three-dimensional dispersion curves in electrically open cases are provided and discussed in order to analyze the influence of gradient coefficient. The distributions of displacement amplitudes are plotted and further applied to investigate the characteristics of complex Rayleigh waves. Finally, the curves of phase speed to the ratio of wave loss decrements are drawn, which provide extra convenience for finding the highest phase velocity points where the complex wave loss near zero. The results of this study can be used as the guidance of experimental measurement of Rayleigh surface waves in FGPM half-space.

## 2. Statement of the Problem

Consider an orthotropic FGPM elastic half-space which contacts with vacuum, and Rayleigh surface wave propagate along the *x*-axis as shown in Figure 1. The horizontal x-y coordinate plane is placed on the top surface of the media, and the *z*-axis of the Cartesian coordinate system is normal to the x-y plane; it is also the poling direction of the piezoelectricity.

For the harmonic wave propagation considered in present paper, the body forces are assumed to be zero; thus, according to the linearity assumption, the dynamic equation for the half-space can be expressed as: (1)$$\begin{array}{c} \nabla \cdot \sigma =\sigma_{ij,j}=\rho {\ddot{u}}_{i}\\ \nabla \cdot D=D_{i,i}=0 \end{array}$$
where the comma subscript denotes a spatial coordinate, the double dot over $u_{i}$ denotes the second derivative with respect to time. The convention that $j\equiv \frac{\partial }{\partial x_{j}}$ is being used, and summation over a repeated subscript is understood. ρ is the mass density of the medium, $u_{i}$ is the mechanical displacement components in the *i*th direction, and σ and D are the mechanical stress tensor and electrical displacement tensor, respectively, which are related to the strain $\varepsilon_{kl}$ and the electric field intensity components $E_{k}$ in the constitutive relations as follows: (2)$$\begin{array}{c} \sigma_{ij}=C_{ijkl}\varepsilon_{kl}-e_{kij}E_{k}\\ D_{j}=e_{jkl}\varepsilon_{kl}+\epsilon_{jk}E_{k} \end{array}$$
where $C_{ijkl}$ is the elastic constants of physical medium, $e_{kij}$ is piezoelectric constants, and $\epsilon_{jk}$ is dielectric constants. Their values vary continuously along the depth direction because the material is made up of FGPM, namely the material properties are the functions of depth in the *z*-axis.

According to elasticity theory, the Cauchy’s strain tensor is $\varepsilon =\frac{1}{2}\left(u\nabla +\nabla u\right)$, and its six components are as follows: (3)$$\varepsilon_{ij}=\frac{1}{2}\left(u_{i,j}+u_{j,i}\right)$$

Based on the quasi-static Maxwell equation, the relationship of the electric field intensity $E_{i}$ and the electrical potential ϕ can be expressed as:(4)$$E_{i}=-\frac{\partial \phi }{\partial x_{i}}$$

In the present paper, only the coupled Rayleigh waves propagating along *x*-axis in the FGPM half-space are considered, and the poling direction is in *z*-direction, thus the mechanical and electrical boundary conditions (electric open case) at the free surface can be written as: (5)$$\begin{array}{ccc} \sigma_{zz} & =0 & \\ \sigma_{xz} & =0 & \left(at\;z=0\right)\\ D_{z} & =0 & \end{array}$$
Now, the problem of Rayleigh wave propagation in the FGPM half-space becomes that of finding the solution of the dynamic Equation (Equation 1) under the boundary condition of Equation (Equation 5).

## 3. Solution of the Problem

For the FGPM half-space polarized in the *z*-direction, the Rayleigh waves propagate along *x*-axis, so all field variables are independent on coordinate *y*. Thus, the governing Equation (Equation 1) in the sagittal plane are given by: (6)$$\begin{array}{l} \frac{\partial \sigma_{xx}}{\partial x}+\frac{\partial \sigma_{xz}}{\partial z}=\rho \frac{\partial^{2}u_{x}}{\partial t^{2}}\\ \frac{\partial \sigma_{xz}}{\partial x}+\frac{\partial \sigma_{zz}}{\partial z}=\rho \frac{\partial^{2}u_{z}}{\partial t^{2}}\\ \frac{\partial D_{x}}{\partial x}+\frac{\partial D_{z}}{\partial z}=0 \end{array}$$

To satisfy the boundary condition mentioned above, the unit step function is introduced here:(7)$$H\left(z\right)=\left\{\begin{array}{cc} 1, & z\geq 0;\\ 0, & elsewhere \end{array}\right.$$
then the mechanical and electric boundary conditions described in Equation (Equation 5) can be automatically incorporated into the constitutive equations of the half-space.
(8)$$\begin{array}{c} \sigma_{xx}=C_{11}\varepsilon_{xx}+C_{13}\varepsilon_{zz}-e_{31}E_{z}\\ \sigma_{xz}=\left(2C_{55}\varepsilon_{xz}-e_{15}E_{x}\right)H\left(z\right)\\ \sigma_{zz}=\left(C_{13}\varepsilon_{xx}+C_{33}\varepsilon_{zz}-e_{33}E_{z}\right)H\left(z\right)\\ D_{x}=2e_{15}\varepsilon_{xz}+\epsilon_{11}E_{x}\\ D_{z}=\left(e_{31}\varepsilon_{xx}+e_{33}\varepsilon_{zz}+\epsilon_{33}E_{z}\right)H\left(z\right) \end{array}$$

For the FGPM half-space in the present paper, whose material properties vary along the positive direction of *z*-axis, the elastic and other material coefficients can be written as: (9)$$\begin{array}{c} \rho (z)=\rho \cdot g_{1}\left(z\right)\\ C(z)=C\cdot g_{2}\left(z\right)\\ e(z)=e\cdot g_{3}\left(z\right)\\ \epsilon (z)=\epsilon \cdot g_{4}\left(z\right) \end{array}$$
to obtain a more universal equation. Here, $g_{1}\left(z\right)$, $g_{2}\left(z\right)$, $g_{3}\left(z\right)$, and $g_{4}\left(z\right)$ are assumed to different gradient functions acting on the mass density ρ, elastic constants of physical medium *C*, piezoelectric constants *e*, and dielectric constants ϵ, respectively.

For a free harmonic half-space surface wave propagation in the *x*-direction, the displacement components and the electric potential can be expressed as:
(10a)$$\begin{array}{cc} u_{x}\left(x,z,t\right) & =e^{i(kx-\omega t)}U\left(z\right) \end{array}$$
(10b)$$\begin{array}{cc} u_{z}\left(x,z,t\right) & =e^{i(kx-\omega t)}W\left(z\right) \end{array}$$
(10c)$$\begin{array}{cc} \phi (x,z,t) & =e^{i(kx-\omega t)}X\left(z\right) \end{array}$$
where U(z) and W(z) represent the amplitudes of vibration corresponding to the directions of *x* and *z*; X(z) represents the amplitude of electric potential; *i* is the imaginary unit;, ω is the angular frequency; and *k* is the complex wave vector in the propagation direction.

Equations (Equation 8)–(10a–c) are inserted into Equation (Equation 6) before dealing with them as the follows: Firstly, swap the two sides terms of the equations for the convenience of expression. Then, reorganize long expressions of every equation based on *k*’s order and pick all the terms of same order with respect to *k* together. Finally, the governing differential equations in terms of displacement components become:
(11a)$$\begin{array}{c} -\rho \omega^{2}H\left(z\right)g_{1}\left(z\right)U\left(z\right)=\\ -g_{2}\left(z\right)C_{11}UH\left(z\right)k^{2}{|}_{k^{2}}^{\mathrm{term}}\\ +[H\left(z\right)\left(g_{2}\left(z\right)\left(C_{13}W^{\prime }+C_{55}W^{\prime }\right)+g_{3}\left(z\right)\left(e_{15}X^{\prime }+e_{31}X^{\prime }\right)\right)\\ +C_{55}W{\left(g_{2}\left(z\right)H\left(z\right)\right)}^{\prime }+e_{15}X{\left(g_{3}\left(z\right)H\left(z\right)\right)}^{\prime }{]ik|}_{k^{1}}^{\mathrm{term}}\\ +\left[H\left(z\right)g_{2}\left(z\right)C_{55}U^{\prime \prime }+{\left(g_{2}\left(z\right)H\left(z\right)\right)}^{\prime }C_{55}U^{\prime }\right]{|}_{k^{0}}^{\mathrm{term}} \end{array}$$
(11b)$$\begin{array}{c} -\rho \omega^{2}H\left(z\right)g_{1}\left(z\right)W\left(z\right)=\\ -\left[g_{2}\left(z\right)C_{55}W+e_{15}g_{3}\left(z\right)X\right]H\left(z\right)k^{2}{|}_{k^{2}}^{\mathrm{term}}\\ +\left[H\left(z\right)g_{2}\left(z\right)\left(C_{13}+C_{55}\right)U^{\prime }+{\left(g_{2}\left(z\right)H\left(z\right)\right)}^{\prime }C_{13}U\right]{ik|}_{k^{1}}^{\mathrm{term}}\\ +[H\left(z\right)\left(g_{2}\left(z\right)C_{33}W^{\prime \prime }+e_{33}g_{3}\left(z\right)X^{\prime \prime }\right)\\ +C_{33}W^{\prime }{\left(g_{2}\left(z\right)H\left(z\right)\right)}^{\prime }+e_{33}X^{\prime }{\left(g_{3}\left(z\right)H\left(z\right)\right)}^{\prime }{]|}_{k^{0}}^{\mathrm{term}} \end{array}$$
(11c)$$\begin{array}{c} -\rho \omega^{2}H\left(z\right)g_{1}\left(z\right)X\left(z\right)=\\ -\left[g_{3}\left(z\right)e_{15}W-\epsilon_{11}g_{4}\left(z\right)X\right]H\left(z\right)k^{2}{|}_{k^{2}}^{\mathrm{term}}\\ +\left[H\left(z\right)g_{3}\left(z\right)\left(e_{15}U^{\prime }+e_{31}U^{\prime }\right)+{\left(g_{3}\left(z\right)H\left(z\right)\right)}^{\prime }e_{31}U\right]{ik|}_{k^{1}}^{\mathrm{term}}\\ +[H\left(z\right)\left(g_{3}\left(z\right)e_{33}W^{\prime \prime }-\epsilon_{33}g_{4}\left(z\right)X^{\prime \prime }\right)\\ +e_{33}W^{\prime }{\left(g_{3}\left(z\right)H\left(z\right)\right)}^{\prime }-\epsilon_{33}X^{\prime }{\left(g_{4}\left(z\right)H\left(z\right)\right)}^{\prime }{]|}_{k^{0}}^{\mathrm{term}} \end{array}$$

In the above equations, the superscripts (${}^{\prime }$) and (${}^{\prime \prime }$) are, respectively, the first and second partial derivatives with respect to *z*. *U*, *W*, and *X* correspond to the U(z), W(z), and X(z) in Equation (10a–c).

To get the solution of Equation (11a–c), U(z), W(z), and X(z) are expanded here into the products of Laguerre orthogonal polynomial series as follows:
(12a)$$\begin{array}{cc} U(z) & =\sum_{m=0}^{\infty }p_{m}^{1}L_{m}\left(z\right) \end{array}$$
(12b)$$\begin{array}{cc} W(z) & =\sum_{m=0}^{\infty }p_{m}^{2}L_{m}\left(z\right) \end{array}$$
(12c)$$\begin{array}{cc} X(z) & =\sum_{m=0}^{\infty }p_{m}^{3}L_{m}\left(z\right) \end{array}$$
where $p_{m}^{i}\left(i=1,2,3\right)$ represent the expansion coefficients which correspond to displacement distributions for wave modes, $L_{m}\left(z\right)=e^{-\frac{z}{2}}P\left(m,z\right)/m!$, and P(m,z) is the *m*th Laguerre polynomial, thus the function $L_{m}\left(z\right)$ forms a complete orthonormal set in the range (0, *∞*), and is therefore suitable for calculating in half-space media. Even though the variable *z* is defined over (0, *∞*), the order of the expansion is always truncated to some value m=M in practice because the solution will converge within a finite number of terms.

Firstly, insert Equation (12a–c) into Equation (11a–c). Then, multiplying both sides of the four equations obtained in the last step by the complex conjugate $L_{j}^{*}\left(z\right)$ with *j* running from 0 to *M*, 3×(M+1) equations appear. Finally, integrating the 3×(M+1) equations over *z* from 0 to infinity, and reordering these equations based on $k^{2}$, $k^{1}$ and $k^{0}$, achieves their matrix representation as follows:(13)$$k^{2}\mathrm{AP}+k\mathrm{BP}+\mathrm{CP}=\omega^{2}\mathrm{MP}$$
where P is a vector composed of $p^{1}$, $p^{2}$, and $p^{3}$ from Equation (12a–c). A, B, C and M are the following matrices, and their exact values can be obtained from Equation (11a–c) based on the coefficients of $k^{2}$, $k^{1}$, and $k^{0}$.
$$\begin{array}{cccc} A & =\left[\begin{array}{ccc} A_{11}^{j,m} & A_{12}^{j,m} & A_{13}^{j,m}\\ A_{21}^{j,m} & A_{22}^{j,m} & A_{23}^{j,m}\\ A_{31}^{j,m} & A_{32}^{j,m} & A_{33}^{j,m} \end{array}\right]\; & B & =\left[\begin{array}{ccc} B_{11}^{j,m} & B_{12}^{j,m} & B_{13}^{j,m}\\ B_{21}^{j,m} & B_{22}^{j,m} & B_{23}^{j,m}\\ B_{31}^{j,m} & B_{32}^{j,m} & B_{33}^{j,m} \end{array}\right]\\ C & =\left[\begin{array}{ccc} C_{11}^{j,m} & C_{12}^{j,m} & C_{13}^{j,m}\\ C_{21}^{j,m} & C_{22}^{j,m} & C_{23}^{j,m}\\ C_{31}^{j,m} & C_{32}^{j,m} & C_{33}^{j,m} \end{array}\right]\; & M & =\left[\begin{array}{ccc} M_{11}^{j,m} & 0 & 0\\ 0 & M_{22}^{j,m} & 0\\ 0 & 0 & M_{33}^{j,m} \end{array}\right] \end{array}$$
$$\;P={\left[p_{0}^{1},p_{1}^{1},\ldots ,p_{M}^{1},p_{0}^{2},p_{1}^{2},\ldots ,p_{M}^{2},p_{0}^{3},p_{1}^{3},\ldots ,p_{M}^{3}\right]}^{T}$$

If only the wave number *k* is considered to be purely real for propagating waves, the solution can be obtained with ease through the conventional approach of the previous study [31]. The conventional method simplifies Equation (Equation 13) as the derivation of Equation (14a–c) in three steps, and Equation (14c) tends to be their final expression, which is a characteristic equation with $\omega^{2}$ as the eigenvalues. Obviously, the method only can resolve the surface wave propagating problem in the real number range, because it takes the wavenumber as the independent variables and the angular frequencies only exist in real number.
(14a)$$\begin{array}{ccc} \overset{\mathrm{NP}}{\overset{︷}{k^{2}\mathrm{AP}+k\mathrm{BP}+\mathrm{CP}}} & = & \overset{\lambda \mathrm{MP}}{\overset{︷}{\omega^{2}\mathrm{MP}}} \end{array}$$
(14b)$$\begin{array}{ccc} \mathrm{NP} & = & \lambda \mathrm{MP} \end{array}$$
(14c)$$\begin{array}{ccc} \mathrm{GP} & = & \lambda P\;(G=M^{-1}N) \end{array}$$

However, in the present paper, the wave number *k* is non-real for complex wave modes, so the process to get the solution becomes more complicated because it entails calculating the propagating and non-propagating wave modes at the same time. Another difficulty lies in the fact that there exists a square term for *k* in the matrix in Equation (Equation 13), which will cause failure to obtain solutions through conventional eigenvalue methods. Thus, it is necessary to reform Equation (Equation 13) to become a classical eigenvalue problem in which the eigenvalue *k* appears linearly.

A new column vector is introduced here:$$\left[\begin{array}{c} q_{m}^{1}\\ q_{m}^{2}\\ q_{m}^{3} \end{array}\right]=k\left[\begin{array}{c} p_{m}^{1}\\ p_{m}^{2}\\ p_{m}^{3} \end{array}\right]\;\left(m=0,1,2,\ldots ,M\right)$$
which can be simplified into
(15)Q=kP

Substituting Equation (Equation 15) into Equation (Equation 13), the result becomes:(16)$$k\mathrm{AQ}+\mathrm{BQ}+\mathrm{CP}=\omega^{2}\mathrm{MP}$$

After reorganizing, the new form of Equation (Equation 16) is
(17)$$A^{-1}\left(C-\omega^{2}M\right)P+A^{-1}\mathrm{BQ}+kQ=0$$

Combining Equations (Equation 15) and (Equation 17) derives
(18)$$\left[\begin{array}{cc} 0 & E\\ A^{-1}\left(\omega^{2}M-C\right) & -A^{-1}B \end{array}\right]R=kR$$
where E is an identity matrix which has the same dimension with $A^{-1}B$, R is a column vector composed of the $p_{m}^{i}$ and $q_{m}^{i}$ from Equation (Equation 15),
$$\begin{array}{cc} R & ={\left[R_{m}^{1}R_{m}^{2}\ldots R_{m}^{6}\right]}^{T}\\ & ={\left[p_{m}^{1}\ldots p_{m}^{3}\;q_{m}^{1}\ldots q_{m}^{3}\right]}^{T}=\left[\begin{array}{c} P\\ Q \end{array}\right] \end{array}$$

The eigenvalues of coefficient matrix of left R in Equation (Equation 18) yield the complex wave number *k*, thereby the phase velocity ω/k, and the eigenvectors allow the calculation of the field distributions. Thus, solving the problem of complex Rayleigh wave propagation becomes calculating the eigenvalue problem of a specified matrix.

## 4. Numerical Results

To demonstrate the influences of the gradient coefficients on the characteristics of surface waves propagating on the free surface of FGPM half-space, some numerical analysis results are given in this section based on the solution derived in the previous section. Even though all the gradient functions $g_{i}\left(z\right)$ may be different because they are bound with different material properties, here, for the convenience of calculation, it is assumed that the variations of the elastic, piezoelectric, and dielectric coefficients and the mass density are the same along the *z*-axis direction, and the FGPM properties are controlled by the same gradient function $g\left(z\right)={\left(1+z\right)}^{n}$, where *n* is the gradient coefficient. Therefore, the material coefficients can be expressed as: $$\begin{array}{c} \rho (z)=\rho \cdot {\left(1+z\right)}^{n}\\ C(z)=C\cdot {\left(1+z\right)}^{n}\\ e(z)=e\cdot {\left(1+z\right)}^{n}\\ \epsilon (z)=\epsilon \cdot {\left(1+z\right)}^{n} \end{array}$$

Lead zirconate titanate (PZT) is a commercially important piezoelectric material, and numerous studies on PZT-based piezoceramics have been conducted by researchers, thus, in the present paper, the set of FGPM half-space is assumed PZT-4, and the material properties (at z=0) for the FGPM half-space are listed in the Table 1.

### 4.1. Approach Validation

To verify the correctness of foregoing formulations, first the displacement curves using the present approach in a non-graded PZT-4 half-space are given, and then a comparison between the results and the published results from the literature [36] is provided. All these displacement curves are normalized with respect to *w* at z=0, and given in Figure 2. Figure 2a was obtained from the present method and Figure 2b was taken from the literature. As can be seen from the comparison, the results from the method proposed in this paper and that from the literature agree very well.

### 4.2. Dispersion Curves for the FGPM Half-Space

For the convenience of understanding the nature of all wave modes and the straightforward visualization of solutions, the dispersion curves are presented in 3D graphics with different colors for clarity. When an angular frequency ω is given based on Equation (Equation 18), a vector is generated which consists of a series of wavenumbers including purely real, purely imaginary, and complex values. After constructing a three-dimensional coordinate system using frequency, purely real values, and purely imaginary values, the dispersion curves are obtained.

The first case is that the gradient coefficient *n* is 0.5 in the gradient function, and the resulting three-dimensional frequency spectra is illustrated in Figure 3. The graphic shows that the pure real (blue dotted curves in Figure 3), imaginary (red dotted curves in Figure 3), and complex roots (green dotted curves in Figure 3) always appear in pairs and are symmetrical about the coordinate plane. This indicates that, if *a* or ai is a solution, then −a or −ai is also a solution, and all complex solutions are evenly distributed in the four quadrants above in the form of conjugates and opposite signs.

It is can be found that the blue curves from the pure real roots represent propagating wave modes, the red curves from the imaginary roots and the green curves from the complex roots correspond to the non-propagating wave modes. For pure real branches, the first branch starts from the cut-off frequency of the first imaginary branch, which is very different from Lamb-like waves starting at zero, but the other real branches’ tendency are similar to Lamb-like waves. As for pure imaginary branches, the first three branches are different from the others because the first branch starts at the frequency zero while ending at the starting point of the real branch, and their next two branches exist between two adjacent cut-off frequencies. Except for the above, the other imaginary branches after the third branch start at a certain cut-off frequency and linked in three-dimensional space by the green complex branches, some of them are coupled together very well. For the green complex branches, they start at frequency 0 and end at the minimum points of the red pure imaginary or the blue pure real branches. It is worth mentioning that the first two complex branches are non-propagating modes at small frequencies and decay more slowly as frequency *f* increase, but their endpoints falls on the blue complex branches. This means the complex branches degenerate into pure real branches, i.e. these non-propagating modes are converted into propagating modes at these points as the complex curves proceed.

To investigate the influence of the heterogeneous character of the FGPM on the dispersion curves, the gradient coefficient *n* is chosen to be 1 and 2 in the present numerical analysis, and a dispersion comparison with *n* is 0.5 is presented. As mentioned above, all these curves are symmetrical about the coordinate plane, thus only the dispersion curves in of one quadrant are given. Figure 4, with the same color scheme as in Figure 3, illustrates two three-dimensional dispersion curves for Rayleigh waves in FGPM half-space with different gradient coefficients. The results in Figure 3 and Figure 4 show that the frequency of the same order propagating mode increases as the gradient coefficient *n* increases for a specified wavenumber. This means that the speed of Rayleigh wave propagation becomes faster as the gradient coefficient *n* increases because the propagation speed of any point on the blue real branches is proportional to the ratio of f/k. For pure imaginary modes, the high order branches tend to become more independent of each other as the gradient coefficient *n* increases, and wave mode conversions happen between them and complex waves. However, the most interesting thing happens with complex waves. Except for those local short complex modes appearing between high order pure imagine modes, the endpoints of complex modes fall on the pure real branches and move forward the *f*-axis gradually as the gradient coefficient *n* increases. As a result, this kind endpoints is reduced by one when *n* is 2, but the other falls on the pure imaginary branch, as illustrated in the Figure 4.

### 4.3. Displacement Amplitude Distributions for the FGPM Half-Space

Next, the displacement amplitude distributions is given to investigate the differences between the propagating waves and the complex Rayleigh waves. Taking the expression $k=k_{R}+ik_{I}\left(k_{R}\in R,k_{I}\in R\right)$ as the conventional solution of the wave number, substituting the expression into Equation (12a–c), and then inserting the result into Equation (10a–c), the new form of displacement can be written as:$$u_{i}\left(x,z,t\right)=\left(\sum_{m=0}^{M}p_{m}^{i}L_{m}\left(z\right)\right)e^{-k_{I}x}*e^{i(k_{R}x-\omega t)}$$

Considering Euler’s formula and assuming t=0, the displacement can be calculated by the following,
(19)$$u_{i}\left(x,z\right)=\left(\sum_{m=0}^{M}p_{m}^{i}L_{m}\left(z\right)\right)e^{-|k_{I}|x}*\cos\left(\left|k_{R}\right|x\right)$$

The displacement field distributions are represented in Figure 5 evaluated at f=22.3825 in the thickness direction (*z*) and the propagation direction (*x*), respectively. Figure 5a shows that the damped exponential distribution happens to the pure imaginary wave modes corresponding to those imaginary wave numbers. The complex Rayleigh wave modes exhibit damped sinusoidal filed distribution with rapid decay, as shown in Figure 5b. This phenomenon becomes more pronounced when the ratio of the real part to the imaginary part in the complex solution is relatively large, as demonstrated in Figure 5c. As the ratio of the real part to the imaginary part becomes larger, the complex wave mode must be converted into pure real wave mode corresponding to propagation wave, as demonstrated in Figure 5d. This consequence is verified through some complex wave branches, such as the curve CA in Figure 4.

### 4.4. The Effects of Gradient Coefficients on the High Phase Velocity Points

To find the highest phase velocity points where the imaginary part of complex wave numbers are near zero is one purpose of the present study. Furthermore, the effects of gradient coefficients on these points is examined. Thus, it is essential to investigate the phase velocity curves of complex wave numbers with different gradient coefficient in the FGPM half-space. Firstly, the real part and imaginary part of complex wave numbers are set to horizontal coordinates that are perpendicular to each other. Then, the phase velocity is set to the vertical coordinates. As a result, a three-dimensional coordinate system about phase velocity and complex wave number is established. The phase velocity curves of complex wave numbers with different gradient coefficient are illustrated in Figure 6, in which the purely imaginary branches are removed to clearly demonstrate the conversion process from complex wave modes to real wave modes. It can be seen clearly that some complex wave modes approach purely real propagating wave modes in this figure, and the phase velocity of the first point which imaginary part near zero becomes large with the increase of gradient coefficient.

To find the exact points where the complex wave modes are near purely real propagating wave modes, the loss ratio η is given. Here, denote the ratio η=Imaginary(k)/Real(k), and set it to the abscissa. Then, set the phase velocity to ordinate, so that the coordinate system is established about loss ratio and phase velocity, as illustrated in Figure 7. As can be seen, the phase velocity of the first conversion point is 6.6 km/s when the gradient coefficient n=1. However, the velocity increases to 7.4 km /s when the gradient coefficient n=2. Thus, it can be concluded that the changes of the gradient coefficients have a great effect on the first point where the imaginary parts of the complex wave modes approach zero, and where the phase velocities of complex waves travel much more quickly than the propagating surface waves. The calculated velocity of propagating surface wave in PZT-4 is 2.2 km/s [36] and the measured value is 2.12 km/s [37].

## 5. Conclusions

A polynomial expansion approach for solving the both propagating and non-propagating Rayleigh modes in the free surface of FGPM half-space has been developed, and the full dispersion spectrum and the displacement amplitude field distributions have been investigated by the present method while presenting all the possible solutions of the pure real, pure imaginary, and the complex. As a result, the following conclusions can be drawn:The improved method proposed in this paper can be used to solve the problem of complex Rayleigh waves in the FGPM half-space. This method can not only provide real roots, but also imaginary and complex roots.The curves of dispersion and displacement demonstrate that some complex wave branches could be converted to the propagation waves.The gradient coefficient *n* has great effect on the frequency of the same branch dispersion curve, and the corresponding frequency will increase with the *n* increase.

## Figures and Tables

**Figure 1 materials-13-02320-f001:**
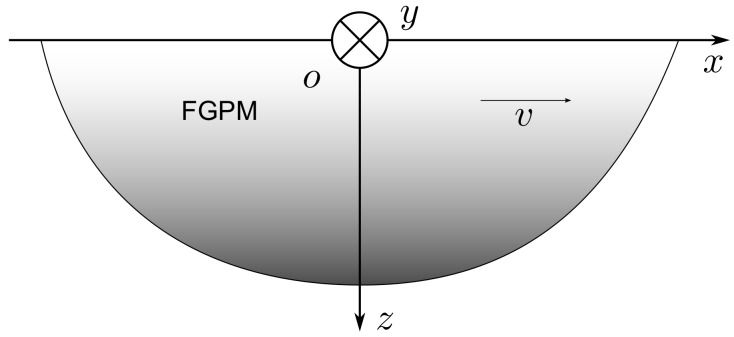
The functionally graded piezoelectric material half-space and coordinate system.

**Figure 2 materials-13-02320-f002:**
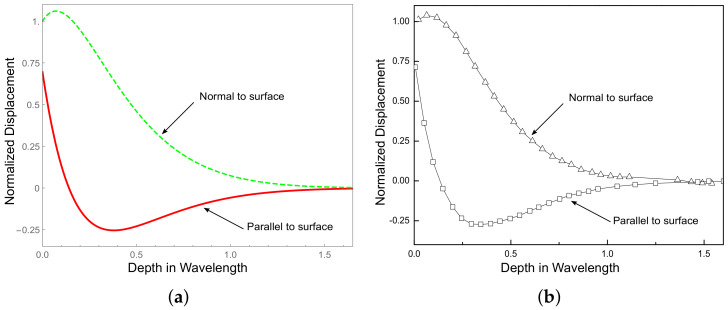
The comparison of displacement curves: (**a**) results from the present method; and (**b**) results from literature.

**Figure 3 materials-13-02320-f003:**
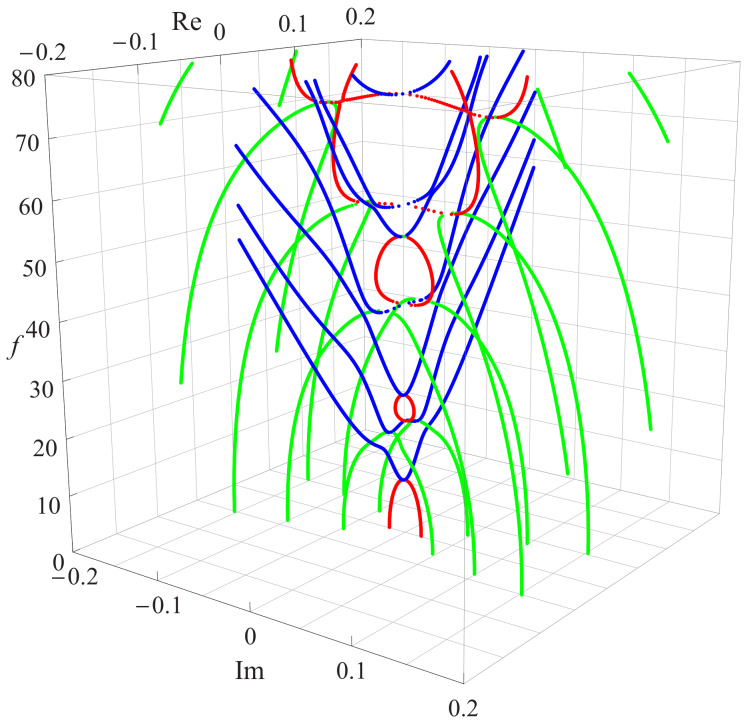
Dispersion curves of complex Rayleigh waves for n=0.5: blue for real, red for imaginary, and green for complex wavenumbers.

**Figure 4 materials-13-02320-f004:**
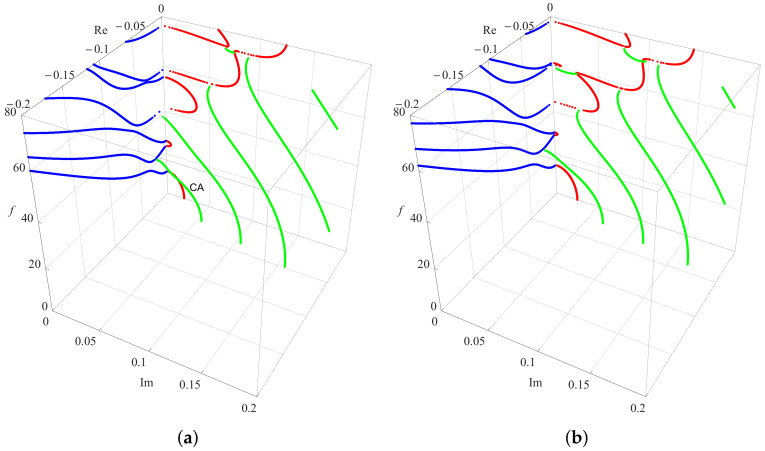
Dispersion curves of complex Rayleigh waves in FGPM half-space: (**a**) n=1; and (**b**) n=2.

**Figure 5 materials-13-02320-f005:**
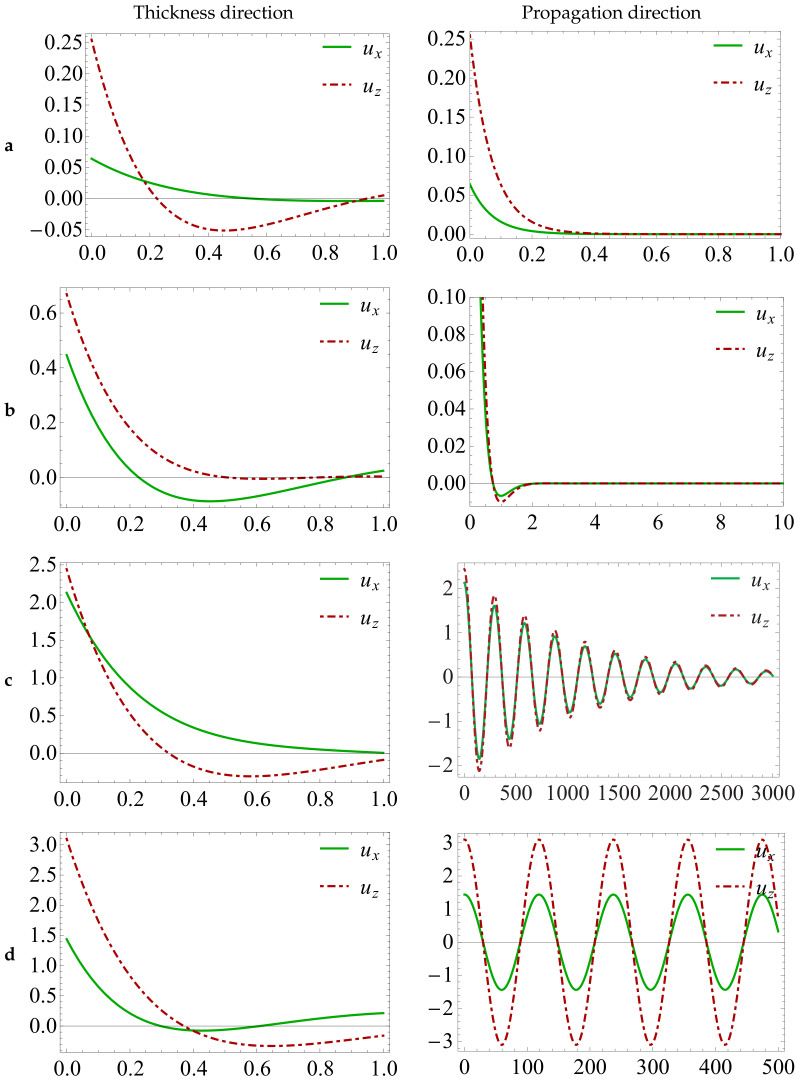
Displacement amplitude field distributions in the thickness *z* and wave propagation *x*-directions for the FGPM half-space, the results got at x=0 and z=0, respectively, with f=22.3825: (**a**) k=13.9275i; (**b**) k=2.1286+3.5681i; (**c**) k=0.0214+0.0010i; and (**d**) k=0.0529.

**Figure 6 materials-13-02320-f006:**
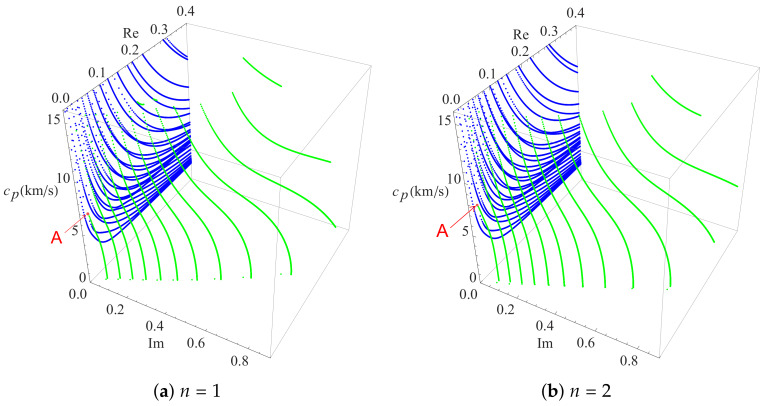
Phase velocity curves of complex Rayleigh waves with different gradient coefficient *n* in FGPM half-space.

**Figure 7 materials-13-02320-f007:**
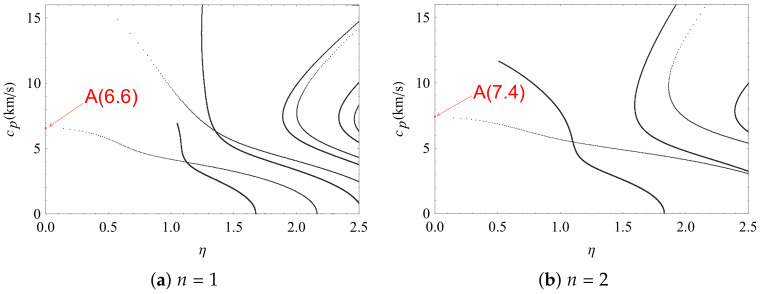
Phase velocity curves of loss decrements with different gradient coefficient *n* in FGPM half-space.

**Table 1 materials-13-02320-t001:** Properties of the piezoelectric PZT-4 [35] at z=0.

Symbols (Unit)	PZT-4
$\rho (kg/m^{3})$	7500
$C_{ij}\left(\mathrm{GPa}\right)$	$\left[\begin{array}{cccccc} 132 & 71 & 73 & 0 & 0 & 0\\ 71 & 132 & 73 & 0 & 0 & 0\\ 73 & 73 & 115 & 0 & 0 & 0\\ 0 & 0 & 0 & 26 & 0 & 0\\ 0 & 0 & 0 & 0 & 26 & 0\\ 0 & 0 & 0 & 0 & 0 & 30.5 \end{array}\right]$
$e_{ij}\left(C/m^{2}\right)$	$\left[\begin{array}{cccccc} 0 & 0 & 0 & 0 & 10.5 & 0\\ 0 & 0 & 0 & 10.5 & 0 & 0\\ -4.1 & -4.1 & 14.1 & 0 & 0 & 0 \end{array}\right]$
$\epsilon_{ij}\left(\times 10^{-9}F/m\right)$	$\left[\begin{array}{ccc} 7.1 & 0 & 0\\ 0 & 7.1 & 0\\ 0 & 0 & 5.8 \end{array}\right]$

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
