# Peer review of "The Complex Rayleigh Waves in a Functionally Graded Piezoelectric Half-Space: An Improvement of the Laguerre Polynomial Approach"

_materials, 2020, doi:10.3390/ma13102320_

Round 1

Reviewer 1 Report

Title: The complex surface waves in a functionally graded piezoelectric half-space: An improvement of the Laguerre polynomial approach

In this paper, the authors used a relatively simpler method called Laguerre polynomial series expansion to calculate the velocity variation of a Rayleigh wave propagating along the thickness direction of PZT4 piezoelectric half space. The results show how the gradient function, volume fraction exponent affect Rayleigh dispersion curves by polynomial numerical simulation. The results are solid and rich enough. However, I think the current version is not suitable for publication in "Materials journal" unless the authors revise the manuscript according to the following suggestions:

Comment 1:

* The work presented in this study combines the Laguerre polynomial method with the solution of the Rayleigh waves propagation problem in terms of FGPM dispersion curves. In the literature, there is a wide variety of studies which have used the FGPM structure and have not be referenced in the current work. For instance:

1/ C. Othmani, F. Takali, A. Njeh, Theoretical study on the dispersion curves of Lamb waves in piezoelectric-semiconductor sandwich plates GaAs-FGPMAlAs: Legendre polynomial series expansion, Superlattices and Microstructures 106 (2017) 86-101”.

2/ C. Othmani, F. Takali, A. Njeh, M. H. Ben ghozlen, Numerical simulation of Lamb waves propagation in a functionally graded piezoelectric plate composed of GaAs-AlAs materials using Legendre polynomial approach, Optik - International Journal for Light and Electron Optics 142 (2017) 401-412.

* In the piezoelectric half-space (Fig. 1) PZT-4 material for the analysis of the current structure is mentioned. Significant research work on this aspect (importance of the PZT4 material) has been ommitted, such as:

3/ C. Othmani, H. Zhang, C.F. Lü, Effects of initial stresses on guided wave propagation in multilayered PZT-4/PZT-5A composites: a polynomial expansion approach, Applied Mathematical Modelling 78 (2020) 14 8–16 8.

Referencing all these studies above is crucial, because it enhances the strength of the proposed manuscript.

- Comment 2:

The electric potential appears in the Eq. 4, but it is not explained. This causes confusion to the reader. It would be useful to add the explanation of the electric potential at that stage rather than later on.

- Comment 3:

We know that the lead role of the unit step function is to apply directly the different mechanical and electric boundary conditions. Accordingly, it will be self-evident to multiplied the mechanical stress and electric potential (Eq. (5)) by the unit step function. So, what is the importance to multiplied Eqs. (7) by this term? I recommend the authors to double check the corresponding equation (Eqs. (7)).

- Comment 4:

Eq. (8.2) is independent of the other two equations. In fact, Eq. (8.2) represents the propagating Love waves in the FGPM half-pace, while Eqs. (8.1) and (8.3) control the propagating Rayleigh waves. As mentioned in section "2. Statement of the problem (Paragraph 1, line 2)", this numerical work just considers the Rayleigh waves, so, please omit the Eq. (8.2). The same thing for the Eq. (9.2), which presents the Love wave propagation equation.

- Comment 5:

Eqs. (9) are developed according the crystallographic direction (polarization according Z-axis direction). So, it is necessary to delete the constants equal a zero, such as C15, C16, C14, C35, C45, C56, e11, e35 in Eq. (9.1). I recommend the authors to double check the corresponding equation (Eqs. (9)). This also allows authors to keep just, the non-zero constants more accurately. The same thing for Eq. (9.3) and (9.4) is required.

- Comment 6:

Just after Eqs. (9): the sentence "In the above equations, the prime (') and double prime ('') represent first and second derivatives with respect to z respectively" should be changed to "where the superscripts (') and (") are, respectively, the first and second partial derivatives with respect to z".

- Comment 7:

Please, delete Eq. (10.2) of the Love wave propagation.

- Comment 8:

Just after Eq. (10): I recommend the authors to double check the equation of Lm(z). I think the correct equation is "exp(-z/2)Lm(z)" (see J. E. Lefebvre, V. Zhang, J. Gazalet, T. Gryba, Conceptual advantages and limitations of the Laguerre polynomial approach to analyze surface acoustic waves in semi-infinite substrates and multilayered structures, J. Appl. Phys. 83, 28 (1998)).

If not, It would be useful to add the explanation of the equation mentioned in the manuscript at that stage rather than later on.

- Comment 9:

How is this amount of eigenmodes [4(M+1)] is generated? Please explain a little bit further, as well as it's necessary to verify properly the validate of [4(M+1)] size of the Rayleigh wave equations.

- Comment 10:

Please check the Eqs. (11) and (12) To be identical consistent, just with Rayleigh waves.

- Comment 11:

Please check the Eqs. (11) and (12) To be identical consistent, just with Rayleigh waves.

- Comment 12:

All the parameters in Eqs. (15) are mentioned above in Eqs. (9), so I suggest to delete these equations, then try to check the next mathematical equations (Eqs. (16), (17)).

- Comment 13:

The sentence "it is assumed that the variations in density, piezoelectric coefficient, dielectric coefficient and elastic coefficient are" should be changed to "it is assumed that the variations of the elastic, piezoelectric, dielectric coefficients and the mass density are".

- Comment 14:

According to Lefebvre et al. (J. E. Lefebvre, V. Zhang, J. Gazalet, T. Gryba, Conceptual advantages and limitations of the Laguerre polynomial approach to analyze surface acoustic waves in semi-infinite substrates and multilayered structures, J. Appl. Phys. 83, 28 (1998)), Laguerre polynomial method can deal with the multilayered plate, only when the material properties of two adjacent layers do not change significantly. So, can you give us idea about the contrast between the different layers of the FGPM half-space?

- Comment 15:

We know that he surface Rayleigh waves is move close to the free surface with very little penetration in the solid medium depth (this observation not goes very well with Laguerre polynomial method). This surface wave is vertically polarized to the surface and it consists of a combination of transversal and longitudinal vibrations in the form of an ellipse, the amplitudes of that depend on the thickness Z under the surface. So, it will be self-evident that the phase velocity of Rayleigh waves in half-space is constant (phase velocity versus frequency is constant=non-dispersion curves). So, can you explain how did you obtained the dispersion curves (Figs. 4, 5, 7 and 8) of Rayleigh waves in the current half-space structure (Fig. 1)?

- Comment 16:

The sentence in conclusions "A new method for solving the propagation and non-propagation waves on" should be changed to "A polynomial expansion approach for solving the both propagating and non-propagating Rayleigh modes in".

- Comment 17:

The English must be carefully revised all over the manuscript.

Author Response

Dear reviewer:

Thanks for your comments and suggestion, which are all valuable and enhances the strength of our manuscript. We’re really grateful to you for the great efforts on reviewing the manuscript. We have carefully addressed all the review comments and improved the quality of this manuscript accordingly. Our responses to recommends are detailed as follows, which also detailed in the revision by highlighting them.

Manuscript ID: materials-759760

Title: The complex surface waves in a functionally graded piezoelectric half-space: An improvement of the Laguerre polynomial approach

Note:

  1. The numbers of equations and figures have been changed because several equations and the figures used to validation have been changed. We are sorry it might cause trouble for your review.
  2. We have also made some improvement to other parts of the manuscript, such as correcting spell errors and grammatical problem. All these are appended at the last page.

Comment 1:

* The work presented in this study combines the Laguerre polynomial method with the solution of the Rayleigh waves propagation problem in terms of FGPM dispersion curves. In the literature, there is a wide variety of studies which have used the FGPM structure and have not be referenced in the current work. For instance:

1/ C. Othmani, F. Takali, A. Njeh, Theoretical study on the dispersion curves of Lamb waves in piezoelectric-semiconductor sandwich plates GaAs-FGPMAlAs: Legendre polynomial series expansion, Superlattices and Microstructures 106 (2017) 86-101”.

2/ C. Othmani, F. Takali, A. Njeh, M. H. Ben ghozlen, Numerical simulation of Lamb waves propagation in a functionally graded piezoelectric plate composed of GaAs-AlAs materials using Legendre polynomial approach, Optik - International Journal for Light and Electron Optics 142 (2017) 401-412.

* In the piezoelectric half-space (Fig. 1) PZT-4 material for the analysis of the current structure is mentioned. Significant research work on this aspect (importance of the PZT4 material) has been ommitted, such as:

3/ C. Othmani, H. Zhang, C.F. Lü, Effects of initial stresses on guided wave propagation in multilayered PZT-4/PZT-5A composites: a polynomial expansion approach, Applied Mathematical Modelling 78 (2020) 14 8–16 8.

Referencing all these studies above is crucial, because it enhances the strength of the proposed manuscript.

Response:

Thank you. These studies can really enhance the strength of the present paper, and have been referenced on the page 2. Reference [15] in the present paper -> 1/ C in your comments, Reference [16] -> 2/ C, and Reference [3] -> 3/ C.

- Comment 2:

The electric potential appears in the Eq. 4, but it is not explained. This causes confusion to the reader. It would be useful to add the explanation of the electric potential at that stage rather than later on.

Response:

Thanks, the explanation for electric potential is given on the upper line of the Eq.4 as follows: (on the page 4)

Based on the quasi-static Maxwell equation, the relationship of the electric field intensity  and the electrical potential  can be expressed as:

- Comment 3:

We know that the lead role of the unit step function is to apply directly the different mechanical and electric boundary conditions. Accordingly, it will be self-evident to multiplied the mechanical stress and electric potential (Eq. (5)) by the unit step function. So, what is the importance to multiplied Eqs. (7) by this term? I recommend the authors to double check the corresponding equation (Eqs. (7)).

Response:

Yes, the unit step function should be multiplied to the Eq.(5), and has be incorporated into the constitutive equations likes the Eq.(8) on the page 5 in the new version, it also be removed from gradient function. Thank you.

- Comment 4:

Eq. (8.2) is independent of the other two equations. In fact, Eq. (8.2) represents the propagating Love waves in the FGPM half-pace, while Eqs. (8.1) and (8.3) control the propagating Rayleigh waves. As mentioned in section "2. Statement of the problem (Paragraph 1, line 2)", this numerical work just considers the Rayleigh waves, so, please omit the Eq. (8.2). The same thing for the Eq. (9.2), which presents the Love wave propagation equation.

Yes, your suggestion makes the complicated simple, and at the same time, makes the theme more prominent. I have revised the Eq.(8) and Eq.(9), and they become the equations (10) and (11) on the page 6 in the new version. Thanks.

- Comment 5:

Eqs. (9) are developed according the crystallographic direction (polarization according Z-axis direction). So, it is necessary to delete the constants equal a zero, such as C15, C16, C14, C35, C45, C56, e11, e35 in Eq. (9.1). I recommend the authors to double check the corresponding equation (Eqs. (9)). This also allows authors to keep just, the non-zero constants more accurately. The same thing for Eq. (9.3) and (9.4) is required.

Response:

Thank you. the constants you mentioned are zero, they should be deleted for the concise and accurate. I have removed them from the related equations. The revised equation is Eq.(11) on the page 6.

- Comment 6:

Just after Eqs. (9): the sentence "In the above equations, the prime (') and double prime ('') represent first and second derivatives with respect to z respectively" should be changed to "where the superscripts (') and (") are, respectively, the first and second partial derivatives with respect to z".

Response:

Thanks, it has been changed in this revision on the page 6.

- Comment 7:

Please, delete Eq. (10.2) of the Love wave propagation.

Response:

Thank you. The Eq.(10.2) has been deleted on the page 7.(now it’s equation 12)

- Comment 8:

Just after Eq. (10): I recommend the authors to double check the equation of . I think the correct equation is "" (see J. E. Lefebvre, V. Zhang, J. Gazalet, T. Gryba, Conceptual advantages and limitations of the Laguerre polynomial approach to analyze surface acoustic waves in semi-infinite substrates and multilayered structures, J. Appl. Phys. 83, 28 (1998)).

If not, it would be useful to add the explanation of the equation mentioned in the manuscript at that stage rather than later on.

Response:

Thanks, the equation ““ mentioned in the literature can obtain right result. However, orthogonal normalization is done in the present manuscript:

So

forms an orthonormal system with respect to the weighted measure function .

In the manuscript, the sentence “ is the th Laguerre polynomial, so the function  forms a complete orthonormal set in the range…” gives explanation about the orthogonal normalization on the page 7 in this version.

- Comment 9:

How is this amount of eigenmodes [4(M+1)] is generated? Please explain a little bit further, as well as it's necessary to verify properly the validate of [4(M+1)] size of the Rayleigh wave equations.

Response:

The variable  in the term  ranges from  to , so multiplying  into any equation in Eqs. 9 will produce  equations. There are four equations in Eqs.9, so the result becomes  equations.

On the page 7 in this revision, it has become  because the equation (9.2) in the  direction has been deleted.

- Comment 10:

Please check the Eqs. (11) and (12) To be identical consistent, just with Rayleigh waves.

Response:

Eq.(11) (now it’s Eq.13 on the page 7) is a simplified form of the previous derivation result. The purpose of Eqs.(12) (now it’s Eq.14 on the page 8) is to illustrate how the conventional method solves the Eq.(11). Eq.(12) shows that the conventional method is to take the matrix  containing the wave number  as the eigenmatrix, and consider  as the eigenvalue, and to solve the angular frequency  by changing the wave number , so it is impossible to give complex waves solution, this is also the reason why the author proposed the improved method.

- Comment 11:

Please check the Eqs. (11) and (12) To be identical consistent, just with Rayleigh waves.

Response:

The author has given response in comment 10.

- Comment 12:

All the parameters in Eqs. (15) are mentioned above in Eqs. (9), so I suggest to delete these equations, then try to check the next mathematical equations (Eqs. (16), (17)).

Response:

Thanks. The Eqs. (15) are really not necessary after the Eqs. (9) (now it’s Eq.11 on the page 6) being revised, and have been deleted. The next mathematical equations have been checked.

- Comment 13:

The sentence "it is assumed that the variations in density, piezoelectric coefficient, dielectric coefficient and elastic coefficient are" should be changed to "it is assumed that the variations of the elastic, piezoelectric, dielectric coefficients and the mass density are".

Response:

Thanks, we have changed the sentence in the first paragraph of the section 4. It becomes: (on the page 9 in this version)

it is assumed that the variations of the elastic, piezoelectric, dielectric coefficients and the mass density are the same along the  axis direction.

- Comment 14:

According to Lefebvre et al. (J. E. Lefebvre, V. Zhang, J. Gazalet, T. Gryba, Conceptual advantages and limitations of the Laguerre polynomial approach to analyze surface acoustic waves in semi-infinite substrates and multilayered structures, J. Appl. Phys. 83, 28 (1998)), Laguerre polynomial method can deal with the multilayered plate, only when the material properties of two adjacent layers do not change significantly. So, can you give us idea about the contrast between the different layers of the FGPM half-space?

Response:

There’re no sudden changes in material properties in the present paper because the gradient coefficients change continuously. However, there exist obvious changes in material properties in the literature you mentioned. That is to say, layered half-space or multi-layered half-space is different from FGPM in structure, so different methods will be used, and will obtain different result. Therefore, we cannot give the contrast between them.

As for the method to solve the waves in multi-layered half-space, we think an approach in an article can be referred (Free-ultrasonic waves in multilayered piezoelectric plates: An improvement of the Legendre polynomial approach for multilayered structures with very dissimilar materials-DOI- 10.1016/j.compositesb.2013.03.024), Maybe the treatment is as follows:

  1. To deal with the cover layer by using the Legendre polynomial;
  2. To address the half-space by using the Laguerre polynomial;
  3. To treat the interface by using the method mentioned in the above literature.

- Comment 15:

We know that the surface Rayleigh waves is move close to the free surface with very little penetration in the solid medium depth (this observation not goes very well with Laguerre polynomial method). This surface wave is vertically polarized to the surface and it consists of a combination of transversal and longitudinal vibrations in the form of an ellipse, the amplitudes of that depend on the thickness Z under the surface. So, it will be self-evident that the phase velocity of Rayleigh waves in half-space is constant (phase velocity versus frequency is constant=non-dispersion curves). So, can you explain how did you obtained the dispersion curves (Figs. 4, 5, 7 and 8) of Rayleigh waves in the current half-space structure (Fig. 1)?

Response:

Yes, Rayleigh waves don’t disperse in an isotropic homogenous half-space. However, the dispersion will happen in an FGPM half-space with material properties varying continuously along z direction. The dispersion curves are obtained as follows:

(1) according to Eq.(17) (it’s Eq.18 in revised version), given a series of angular frequency values , a series of corresponding wave numbers will be generated in the form of eigenvalues.

(2)  will give  which always is real. As for wave numbers, they can be classified into three types according to the purely real, the pure imaginary and the complex.

(3) Taking the real wavenumbers as the -axis, imaginary wavenumbers as the -axis, and  as the z-axis, the Figs. 4, 5, 7 and 8 will be given. (Now they are figures 3,4,6,7)

- Comment 16:

The sentence in conclusions "A new method for solving the propagation and non-propagation waves on" should be changed to "A polynomial expansion approach for solving the both propagating and non-propagating Rayleigh modes in".

Response:

Thanks, the sentence has been revised based on your comment (on the page 15).

- Comment 17:

The English must be carefully revised all over the manuscript.

Response:

Thanks, the author has checked the English in the manuscript, and revised several grammar and expression issues.

Other modifications:

  1. complex surface waves -> complex Rayleigh waves

Reason: (1) the second reviewer’s comment 2.1, (2) only investigated in x and z direction

  1. the word: image -> imaginary on the page 3, 12, 14, 15.

Reason: error in typed

  1. Remove the text “PZT-4” in Fig. 1

Reason: PZT-4 is just a material used in example, shouldn’t appear in the figure.

  1. Inserting “at “ in equation (5) makes itself more complete.
  2. The sentence “In the case of wave number k is real for propagating waves” on the page 7 of this version revised to “If only the wave number k is considered to be purely real for propagating waves”.

Reason: to express more accurately.

  1. The reasons for using PZT materials are given on the page 9 of this version, at the same time, modified the expression of the material properties table to make the expression clearer.
  2. Correcting the writing of the equation (19) on the page 12 of this version.

Reviewer 2 Report

  1. Some formal improvements are needed:
    1. at moving from Chapter 2 to Chapter 3 the notation of material properties is changed from tensor to matrix form, but without explanation - this should be added!
    2. at the beginning of Chapter 4 the material coefficients are given as functions of coordinate z, but differing from (7) regarding the use of step function H(z). Maybe, this has no significance but it shoud be commented.
    3. Figure 4 exhibits colored curves assigned to different types of solutions. In the text below the Figure the assignments red and blue are reversed.
  2. Remarks and questions:
    1. Many times the terms 'surface waves' and 'Rayleigh waves' are used in this articel, obviously also in situations of doubtfulness if the usual understanding of these types of acoustic phenomena at a stress-free surface of a half-space is fulfilled. For example, in the special case of Figure 4 (non-graded material) one would expect only the well-known sagitally polarized surface wave - the Rayleigh wave. The question arises what type of additional waves have also a real wave number as contained in Figure 4. And all other waves including pure imaginary roots of wave number 'k' are also named 'Rayleigh surface waves'. It seems there are comments and/or clearly different names required.
    2. In this context, the authors should be also clear about the history of correctness to distinguish between different types of complex surface waves (in the literature often considerered as PSAW = pseudo surface acoustic waves, especially HVPSAW = high velocity ... because of practical importance). In the current paper are literature citations [17] and [24] with HVPSAW discussions for LiNbO3 with Euler angles (0,-49,0). But they cannot excited by surface sources (see V. Biryukov and M. Weihnacht, Real-space field of surface sources and the problem of fast leaky wave generation in a piezoelectric half-space, J. Appl. Phys. 83 (1998) 3276-3287). Such solutions should be ruled out as HVPSAW and better considered as bulk wave reflection situations.
    3. At the end of Chapter 4.4 is formulated: "...phase velocity much faster than that of bulk waves..". The question is for people who are not familiar with acoustics in FGM and FGPM what is the definition of bulk waves in such material systems and how to calculate them.
    4. Why was chosen for the Approach validation an aluminum half-space and not a non-graded PZT-4 variant. And which is the material used for Figures 4 and 5?
    5. What are the selection criteria for the cases a, b, c, d in Figure 6? In the sense of the above discussion of 1. and 2. it would be important to show that also for all other k-solutions at given frequency the displacement amplitude decreases in thickness direction. If not, comments should be given.

Author Response

Dear reviewer:

Thanks for your comments and suggestion, which are all valuable and enhances the strength of our manuscript. We’re really grateful to you for the great efforts on reviewing the manuscript. We have carefully addressed all the review comments and improved the quality of this manuscript accordingly. Our responses to recommends are detailed as follows, which also detailed in the revision by highlighting them.

Manuscript ID: materials-759760

Title: The complex surface waves in a functionally graded piezoelectric half-space: An improvement of the Laguerre polynomial approach

Note:

  1. The numbers of equations and figures have been changed because several equations and the figures used to validation have been changed. We are sorry it might cause trouble for your review.
  2. We have also made some improvement to other parts of the manuscript, such as correcting spell errors and grammatical problem. All these are appended at the last page.

Comments and Suggestions for Authors

  1. Some formal improvements are needed:

(1).  at moving from Chapter 2 to Chapter 3 the notation of material properties is changed from tensor to matrix form, but without explanation - this should be added!

Response:

Thank you. The tensor has been expanded and explanation has been given in the revised edition. (on the page 5 in this version)

(2).  at the beginning of Chapter 4 the material coefficients are given as functions of coordinate z, but differing from (7) regarding the use of step function H(z). Maybe, this has no significance but it should be commented.

Response:

Thank you, the  in the equations (7) is not necessary any more in the new version, and has been removed, because the modification on the page 5.

(3).  Figure 4 exhibits colored curves assigned to different types of solutions. In the text below the Figure the assignments red and blue are reversed.

Response:

Thanks, you are right. The two errors in the text have been corrected and the description become consistent with the figure 4. (on the pages 10 and 11 in this version)

  1. Remarks and questions:

(1).  Many times the terms 'surface waves' and 'Rayleigh waves' are used in this article, obviously also in situations of doubtfulness if the usual understanding of these types of acoustic phenomena at a stress-free surface of a half-space is fulfilled. For example, in the special case of Figure 4 (non-graded material) one would expect only the well-known sagitally polarized surface wave - the Rayleigh wave. The question arises what type of additional waves have also a real wave number as contained in Figure 4. And all other waves including pure imaginary roots of wave number 'k' are also named 'Rayleigh surface waves'. It seems there are comments and/or clearly different names required.

Response:

Thanks for your comment.

The gradient coefficient in Figure 4 (now it’s figure 3) should be 0.5, which has been corrected in this version.

The present paper aims to give all possible solutions of wavenumber . We all know that, the purely real wavenumbers indicate propagating waves, namely, Rayleigh surface waves. As for the purely imaginary and complex roots, they refer to non-propagating waves, and their displacements will decrease with the thickness increases, namely, they are parts of Rayleigh-like surface waves. So, the purely real waves, the purely imaginary waves and the complex waves, they all are complex Rayleigh surface waves.

We have made some revision based on your suggestion, for example, correcting the captions of the Figures (3), (4) and (6).

(2).  In this context, the authors should be also clear about the history of correctness to distinguish between different types of complex surface waves (in the literature often considered as PSAW = pseudo surface acoustic waves, especially HVPSAW = high velocity ... because of practical importance). In the current paper are literature citations [17] and [24] with HVPSAW discussions for LiNbO3 with Euler angles (0,-49,0). But they cannot excited by surface sources (see V. Biryukov and M. Weihnacht, Real-space field of surface sources and the problem of fast leaky wave generation in a piezoelectric half-space, J. Appl. Phys. 83 (1998) 3276-3287). Such solutions should be ruled out as HVPSAW and better considered as bulk wave reflection situations.

Response:

As you said, the solutions really should be ruled out as HVPSAW based on your mentioned literature from 1998. We have deleted the literature citations [17] and [24], and replaced with others. Thank you very much. (on the pages 2 and 3 in this version)

(3).  At the end of Chapter 4.4 is formulated: "...phase velocity much faster than that of bulk waves..". The question is for people who are not familiar with acoustics in FGM and FGPM what is the definition of bulk waves in such material systems and how to calculate them.

Response:

Thanks. Here exists a problem that the expression is not accurate enough, which has been revised to:

… phase velocities of complex waves travel are much faster than that of propagating surface waves can do. The calculated velocity of propagating surface wave in PZT-4 is  [36], and the measured value is  [37]. (on the page 15 in this version)

(4).  Why was chosen for the Approach validation an aluminum half-space and not a non-graded PZT-4 variant. And which is the material used for Figures 4 and 5?

Response:

The reason why Aluminum was chosen for validation is that the compared curves in the literature is the clearest and obtained from a half-space.

In order to unify the materials used in all the examples of this manuscript, we have replaced the material with PZT-4. Now, all materials used in this manuscript are PZT-4.

Thank you.

(5).  What are the selection criteria for the cases a, b, c, d in Figure 6? In the sense of the above discussion of 1. and 2. it would be important to show that also for all other k-solutions at given frequency the displacement amplitude decreases in thickness direction. If not, comments should be given.

Response:

Thanks for your comments. The subfigures a, b, c and d were selected as examples because they are very representative, which showed 4 typical situations. From these figures we can see that: figure a represents the case of purely imaginary, figure b is a typical example of complex wavenumbers, figure c illustrates that the wave attenuation gradually slows down with the imaginary part of the complex number decreases, figure d describes a propagation situation when the wavenumber is purely real.

As for all other k-solutions, their displacement amplitudes also decrease in thickness direction at given frequency. The following equations give the reason:

The displacement equation is

and for where  

is the Laguerre polynomial, so after arranging the equation, it becomes

We can see from the equation: when the , ,  and  are fixed values,  will becomes smaller overall with the  increases due to the existence of . That is to say, the displacement amplitudes decrease in thickness direction at given frequency for all other k-solutions.

Other modifications:

  1. complex surface waves -> complex Rayleigh waves

Reason: (1) the second reviewer’s comment 2.1, (2) only investigated in x and z direction

  1. the word: image -> imaginary on the page 3, 12, 14, 15.

Reason: error in typed

  1. Remove the text “PZT-4” in Fig. 1

Reason: PZT-4 is just a material used in example, shouldn’t appear in the figure.

  1. Inserting “at “ in equation (5) makes itself more complete.
  2. The sentence “In the case of wave number k is real for propagating waves” on the page 7 of this version revised to “If only the wave number k is considered to be purely real for propagating waves”.

Reason: to express more accurately.

  1. The reasons for using PZT materials are given on the page 9 of this version, at the same time, modified the expression of the material properties table to make the expression clearer.
  2. Correcting the writing of the equation (19) on the page 12 of this version.

Round 2

Reviewer 1 Report

The paper has been revised according to the comments and the revision is satisfactory. It can be accepted for publication in the journal as is.